# Exploring Mosquito Excreta as an Alternative Sample Type for Improving Arbovirus Surveillance in Australia

**DOI:** 10.3390/pathogens14010042

**Published:** 2025-01-08

**Authors:** Tess R. Malcolm, Melissa J. Klein, Karolina Petkovic, Ina Smith, Kim R. Blasdell

**Affiliations:** 1Health and Biosecurity, Commonwealth Scientific and Industrial Research Organisation, Geelong, VIC 3220, Australia; melissa.klein@csiro.au; 2Manufacturing, Commonwealth Scientific and Industrial Research Organisation, Clayton, VIC 3168, Australia; karolina.petkovic@csiro.au; 3Health and Biosecurity, Commonwealth Scientific and Industrial Research Organisation, Canberra, ACT 2601, Australia; ina.smith@csiro.au

**Keywords:** arbovirus, surveillance, mosquito excreta, magnetic virus concentration, Dengue virus

## Abstract

Current arbovirus surveillance strategies in Australia involve mosquito collection, species identification, and virus detection. These processes are labour-intensive, expensive, and time-consuming and can lead to delays in reporting. Mosquito excreta has been proposed as an alternative sample type to whole mosquito collection, with potential to streamline the virus surveillance pipeline. In this study, we investigated the feasibility of *Aedes aegypti* excreta as a sample type in the detection of Dengue virus serotype 2 (DENV2). DENV2 could be detected from as little as one DENV2-infected mosquito excreta spot, with virus levels in individual excreta spots varying within and between mosquitoes and depending highly on mosquito viral load. Detectability was improved by pooling up to 20 DENV2-infected mosquitoes and collecting excreta into liquid substrate, followed by virus concentration using magnetic nanoparticles. Virus concentration improves quantification accuracy in comparison to unconcentrated samples and increases the amount of material available for detection, expanding detection capabilities to techniques with higher limits of detection. Mosquito excreta as a sample type, coupled with magnetic virus concentration, expands the current detection toolbox for DENV2 and has the potential to improve arbovirus surveillance strategies in Australia.

## 1. Introduction

Over 3.9 billion people worldwide living in tropical and sub-tropical regions are at risk of arthropod-borne virus (arbovirus) disease, such as dengue fever, yellow fever, Zika, and chikungunya fever [1]. Dengue virus (DENV) alone causes over 96 million symptomatic cases and approximately 40,000 deaths annually [2], making it the most prevalent arbovirus transmitted by *Aedes* mosquitoes. Arboviruses pose not only a health risk but also an economic risk to those in transmission areas, with the financial burden from these diseases significantly impacting local economies. Arboviruses transmitted by *Aedes aegypti* and *Ae. albopictus* have an estimated annual global financial burden of USD 3.1 billion, a figure that peaked at USD 20.3 billion in 2013 [3]. Management and prevention strategies, including surveillance, receive ongoing investment at only a fraction of the amount spent on damages, with the global cost estimated at USD 166.3 million [3].

In Australia, arbovirus surveillance is conducted via a combination of mosquito trapping and sentinel animal monitoring and is managed individually by each state or territory [4,5]. Sentinel chicken populations are maintained in New South Wales, Northern Territory, South Australia, and Western Australia and are monitored for seroconversion to Murray Valley Encephalitis Virus (MVEV), Japanese Encephalitis Virus (JEV), and West Nile Virus (WNV) Kunjin strain [6,7,8,9,10]. Despite a decades-long reliance on the early warning system provided by sentinel chickens, states are moving away from this surveillance method to circumvent issues with cost, labour, unreliability, and animal ethics [4]. The alternative method of mosquito trapping occurs ubiquitously across Australia. Trapping and collection of mosquitoes are coordinated by councils identified as at risk of arbovirus transmission and involve the strategic placement of mosquito traps followed by the collection and shipping of mosquitoes to a designated diagnostic laboratory in the state capital. Here, mosquito populations are laboriously identified to species level and counted, followed by viral RNA detection by the PCR assay of pools consisting of up to 1000 mosquitoes to detect for MVEV, JEV, WNV (Kunjin), Ross River Virus (RRV), and Barmah Forest Virus (BFV). Arbovirus monitoring may not be carried out by all states or councils involved in mosquito trapping, and if executed, it is highly dependent on local resource availability. The entire process, from setting of traps to collection and laboratory analysis, is time-consuming, labour-intensive, and expensive, all of which often culminate in delays to reporting. The 2022 JEV outbreak is an example of how the currently incohesive approach to arbovirus surveillance in Australia can result in outbreaks spreading undetected. JEV was first detected in piggeries and through human case diagnoses and was only detected from archived mosquito surveillance samples collected during the 2021–2022 season after the outbreak was established throughout Queensland, New South Wales, Victoria, and South Australia [11]. The lack of early warning in this case is likely attributed to geographic coverage, highly specific molecular testing, and the overarching disparities in objectives, targeted viruses, methodologies, funding, data reporting, and data sharing between neighbouring regions [11]. The 2022 Australian JEV outbreak highlighted the need for an updated and consistent model of surveillance throughout Australia, underpinned by novel detection techniques that demand fewer resources than those currently available.

Recent advances in novel detection techniques have predominantly come in the form of exploring alternative sample types to whole mosquitoes. Honey-baited Flinders Technology Associates (FTA) RNA preservation cards have been explored as an alternative sample collection method, relying on the expectoration of saliva containing virus onto cards during feeding [12,13,14,15,16,17]. Cards can then be collected and posted to diagnostic laboratories for PCR analysis without the need for cold storage or homogenisation of large mosquito pools. Although a promising sample type in terms of logistics, mosquito saliva is deposited in such small amounts (~4.7 nl) that accurate virus detection is often compromised [18]. More recently, mosquito excreta has been identified as a potential sample type for surveillance of circulating arboviruses. Fontaine and coworkers (2016) investigated Dengue virus serotype 1 (DENV1) excretion in the context of viral dissemination throughout *Aedes* mosquitoes, finding that DENV1 detection in excreta could predict systemic viral dissemination [19]. DENV2 was similarly reported as detectable in *Ae. aegypti* mosquito excreta using immunoassay techniques [20]. WNV and RRV can be detected in the excreta of *Culex annulirostris* and *Ae. vigilax*, respectively, as the primary vectors of these viruses in Australia [21,22]. WNV and RRV, along with Murray Valley Encephalitis Virus (MVEV), have further been detected from field-collected excreta samples using modified light traps and excreta collected onto FTA cards to collect samples [23]. Excreta presents itself as an attractive sample type due to its ease of collection, volume of material [24], and the ability to monitor virus infection over time in a way that is non-destructive to mosquitoes.

This study evaluated the detectability of Dengue virus serotype 2 (DENV2) in excreta from infected *Ae. aegypti* mosquitoes and assessed the application of excreta as a potential sample type for next-generation surveillance strategies. We investigated collection logistics and method optimisation for optimal retrieval of viruses and found that excreta collection into a liquid substrate coupled with magnetic concentration enhanced detection capabilities. With this study, we demonstrate the suitability of mosquito excreta for use as a viral detection sample and provide a framework to further improve on the sample collection and preparation pipelines required for arbovirus detection from mosquito-derived samples.

## 2. Materials and Methods

### 2.1. Dengue Virus Serotype 2 Culture and Isolation

Dengue virus type 2 (DENV2) isolate ET300, isolated from a soldier returning from East Timor, was used for experiments. DENV2 was passaged in C6/36 cells (*Aedes albopictus*) and Vero cells (African green monkey kidney). The resulting viral supernatant was clarified and titrated in Vero cells, with the titre calculated using the Reed Muench method [25]. DENV2 viral supernatant aliquots were frozen at −80 °C until use.

### 2.2. Breeding, Raising, and Infecting Aedes aegypti

*Ae. aegypti* mosquitoes were housed at 26.5 °C, 65% relative humidity, and a 12:12 h light/dark cycle and provided with a 10% *w*/*v* sucrose solution ad libitum. Five-to-ten-day-old female mosquitoes were starved of sucrose solution overnight before being challenged with DENV2-spiked human blood meal (TCID_50_ 10^6^/mL final concentration) (Lifeblood 23-11VIC07) using a Hemotek blood feeder device and collagen membrane (Hemotek, Blackburn, UK). After a 1-h feeding period, engorged females were transferred to a new cage and housed for 7 days with maintenance conditions as above. At 7 days post infection (dpi), mosquitoes were anaesthetised by freezing for 90 s and intermittently exposed to carbon dioxide (CO_2_), and mosquitoes were selected at random and placed in paper cups designed for excreta collection onto a solid surface (dissolvable paper) or into a liquid substrate (PBS-A; 137 mM NaCl, 2.7 mM KCl, 8 mM Na_2_HPO_4_, and 2 mM KH_2_HPO_4_, pH 7.4). Mosquitoes were provided with paper towels soaked with 10% *w*/*v* sucrose solution placed on top of the cup. The sucrose solution was spiked with blue food dye according to experimental requirements. At 10 dpi, whole mosquitoes were destroyed by freezing for 20 min, then collected into 200 µL Qiagen RLT Plus buffer (Qiagen, Hilden, Germany) and stored at −80 °C until use. Dissolvable paper, or PBS-A, was collected from each cup and stored in individual plastic bags or tubes at −80 °C until further use.

### 2.3. RNA Extraction, cDNA Generation, and qPCR

Whole mosquitoes were added to a tube containing five chrome steel beads (2.3 mm; Daintree Scientific Australia, St. Helens, Australia) in a total volume of 350 µL RLT Plus buffer and homogenised using a Qiagen Tissue Lyser (Qiagen) with the following program: 1 min at 30 oscillations/s and 3 min at 30 oscillations/s. RNA was extracted using the RNeasy Plus Mini Kit (Qiagen) according to the manufacturer’s instructions and stored at −80 °C until cDNA generation. cDNA was generated using the SuperScript^TM^ III First-Strand Synthesis System (Thermo Fisher Scientific, Waltham, MA, USA), using 7 µL of extracted RNA and random hexamers according to the manufacturer’s instructions. cDNA was stored at −20 °C until further use. DENV2 infection status of individual mosquitoes was evaluated using quantitative PCR (qPCR). DENV non-structural protein 1 (NS1) gene-specific primers (Forward: 5′-ACGTGCACACATGGACAGA-3′, Reverse: 5′-ACTGAGCGGATTCCACAAA-3′) at 0.2 µM, together with TB Green© Premix Ex Taq^TM^ II (Tli RNase H Plus; Takara Scientifix Pty Ltd., Shiga, Japan), were used, and qPCR was carried out in technical duplicate using the MicroAmp^TM^ Fast Optical 96-well Reaction Plate with Barcode, 0.1 mL (Applied Biosystems, Thermo Fisher Scientific, Waltham, MA, USA). qPCR was performed using a QuantStudio^TM^ 6 instrument (Applied Biosystems) and the following cycling conditions: 30 s activation phase at 95.0 °C, 40 cycles of 5 s at 95.0 °C followed by 30 s at 60.0 °C. Ct values were automatically calculated using the QuantStudio^TM^ Real-Time PCR Software (Applied Biosystems) with a Ct threshold of 0.2. Technical duplicates were used to calculate the average Ct value of each mosquito, and mosquitoes were deemed infected if the average Ct value was below 35.0. Mosquitoes were grouped into ‘infected’ and ‘not infected’ pools, and the number of excreta spots per mosquito was plotted using Prism GraphPad (v9.1.2). Prism GraphPad was used to perform a two-tailed *t*-test to determine any statistically significant differences in excretion frequency between the two groups.

### 2.4. Detection of DENV2 in Mosquito Excreta Using qPCR

Mosquito excreta was collected from 50 mosquitoes between 7 and 10 dpi on dissolvable paper discs lining the bases of cups of individually housed mosquitoes (Figure 1a). A lightweight fabric work stabiliser (Sulky, Solvy water-soluble stabiliser, Kennesaw, GA, USA), with comparable thickness to printer paper, was used to create the dissolvable paper discs. Mosquitoes were provided with a 10% *w*/*v* sucrose solution spiked with blue food dye, resulting in excreta being deposited on the dissolvable paper as blue spots. Spots were photographed, counted by eye, and visually evaluated. Individual spots were excised from paper discs using a 2 mm round hole punch. Hole-punched spots were dissolved in 350 µL of RLT Plus buffer (Qiagen) in batches of 1, 5, 10, 20, or 50 spots according to the experiment. For single spot experiments, 10 spots from a single mosquito were excised and processed individually as 10 separate replicates. For batches of 5 and 10, excreta spots were excised from a single mosquito and pooled into three replicate pools of 5 and three replicate pools of 10. For batch sizes of 20 and 50, excreta spots were excised in equal numbers from five different mosquitoes and pooled into three replicate pools of 20 spots and three replicate pools of 50 spots. Excreta samples for these experiments were selected based on mosquito viral load and number of spots per disc. RNA extraction, cDNA generation, and qPCR analyses were performed in technical duplicate as described above. Each mosquito excreta sample was tested using qPCR in technical duplicate as described above and deemed positive if Ct values were below 36. A higher Ct cut-off was selected compared to the whole mosquito experiments to reflect the reduction in material used for testing.

### 2.5. Mosquito Excreta Collection into Liquid Substrate

Mosquitoes were bred, raised, and infected as described above. At 7 dpi, pools of 5, 10, or 20 mosquitoes were placed into cups with a mesh bottom insert, with each pool repeated in triplicate. Cups were placed within a sample container filled with 5 mL of PBS-A and supplied with 10% *w*/*v* sucrose solution ad libitum (Figure 2a). Excreta was collected into the PBS-A from 7 to 10 dpi. At the conclusion of the experiment, mosquitoes were destroyed as described above, and the PBS-A containing the excreta was stored at −80 °C until use. Mosquitoes were homogenised in experimental pools, except for 20-mosquito pools, which were divided into two 10-mosquito pools to aid in the homogenisation process and recombined following homogenisation and RNA extraction. RNA extraction, cDNA generation, and qPCR were performed as described above.

### 2.6. Concentration and Detection of DENV2 in Mosquito Excreta Using Magnetic Beads

Prior to concentration with magnetic beads, 100 µL was aliquoted from each excreta/PBS-A sample and kept as the ‘unconcentrated’ control for comparison against the magnetically concentrated sample. Mag4C-Lv magnetic beads (50 µL, OZ Biosciences) were added to the remaining excreta/PBS-A samples and incubated for 1 h at room temperature with rocking. Samples were then placed onto a magnetic separation rack for 20 min at room temperature to allow collection of the magnetic beads and captured virus. Supernatant was removed, and 100 µL of elution buffer (OZ Biosciences) was used to resuspend beads and virus. Samples were gently vortexed for 10 min before RNA extraction, cDNA generation, and qPCR as described above. cDNA copies/µL were calculated using a standard curve generated using 10-fold dilutions of DENV2 cDNA with known copy number/µL.

## 3. Results

### 3.1. Mosquito Excretion Patterns Are Not Correlated with Infection Status

*Ae. aegypti* mosquitoes were successfully infected with Dengue virus serotype 2 with an infection rate of 66% as expected for this strain and experimental conditions used [26]. Thirty of the 33 infected mosquitoes recorded Ct values below 22, while only three were at the upper limit of the positive range. A further four mosquitoes recorded Ct values above 35 and were counted as uninfected for the purposes of this experiment. All mosquitoes remained healthy throughout the 7–10 dpi period in which mosquitoes were housed individually, except for one that died between days 9 and 10. The excretion patterns of individual mosquitoes were monitored using sugar solution tinted with blood food dye, which produces blue-tinted excreta spots for ease of observation and counting (Figure 1b). Food-safe dyes are not toxic to mosquitoes and have been shown to not impact feeding behaviours when used in food intake tracking studies [27]. Blue excreta collected onto dissolvable paper discs had appearances ranging from bright blue spots with defined edges to weak blue patches with undefined edges (Figure 1b). There were no observable visual differences between infected and uninfected mosquitoes, nor any correlation between viral load and excreta appearance. The number of excreta spots per disc varied from 0 to 110, with an average ± standard error of the mean (SEM) of 34.0 ± 3.1 spots/disc, or 11.3 ± 1.0 excreta spots a day/mosquito. When grouped into infected (Ct < 35.0) vs. uninfected (Ct > 35.0), average spots/disc ± SEM were 33.2 ± 4.1 and 35.4 ± 4.5, respectively. Statistical analyses (two-tailed *t*-test) indicated no statistically significant difference in excretion frequency due to infection status (Figure 1c).

### 3.2. DENV2 Can Be Detected from a Single Mosquito Excreta Spot

Following the collection of excreta, we investigated the lowest number of excreta spots required to detect DENV2 using qPCR and identified the average percentage of DENV2-positive excreta spots from a single mosquito. Single excreta spots were excised from dissolvable paper discs and dissolved in RLT Plus buffer before RNA extraction and cDNA generation for qPCR. Spots were tested in technical duplicates and were considered positive if Ct scores of both duplicates were below 36.0. A slightly higher Ct cut-off was used for these experiments, as viral load in a single excreta spot was expected to be much lower than that observed from a whole mosquito. Excreta spots with only one of the technical duplicates recording a Ct score were designated as equivocal. When spots did not meet the outlined criteria, they were deemed negative. Five mosquitoes with a Ct score within a 0.69 range (17.03–17.72), that each excreted between 10 and 15 spots, were selected for comparison (Table 1). These mosquitoes were selected based on high viral load and feasibility to test all excreta spots deposited during the experiment.

The number of positive excreta spots varied between mosquitoes. All 10 excreta spots from mosquitoes 5 and 28 tested positive or equivocal in qPCR experiments (Table 1). Mosquitoes 44 and 42 recorded six and five positive spots, respectively, including those deemed equivocal, while just one excreta spot from mosquito 1 tested positive. Excreta Ct scores also varied between spots from the same mosquito. Mosquito 5 excreta Ct scores ranged from 27.45 to 34.79 with an average score of 30.29. Similarly, mosquito 44 excreta Ct scores ranged from 27.10 to 34.22 with an average of 32.39. These ranges indicate viral load can differ by ~100-fold between individual spots from a single mosquito and that virus detection from excreta is highly dependent on individual spot collection.

### 3.3. DENV2 Detection in Excreta Is Dependent on Mosquito Viral Load

The high variability in viral load between both excreta spots from a single mosquito and excreta spots from different mosquitoes means that viral detection is highly dependent on individual spot selection. To eliminate the need to test every excreta spot, we tested pooling excreta to improve the likelihood of virus detection. Individual mosquitoes were selected based on whole mosquito Ct value and number of excreta spots per disc. For this experiment, spots from six individual mosquitoes were pooled into batches of 5 or 10 and dissolved in RLT Plus buffer (Qiagen) for RNA extraction followed by cDNA generation and qPCR analysis.

DENV2 detection in excreta batches was highly dependent on whole mosquito viral load. Detection appeared to peak in excreta from mosquitoes with Ct scores between 17 and 18, with mosquito 21 (Ct = 17.52) and mosquito 19 (Ct = 17.87) excreta testing positive in all or almost all pools (Table 2). Interestingly, mosquito 4 (Ct = 16.72), which had the lowest whole mosquito Ct score of all mosquitoes tested, had positive hits from all 5-excreta spot pools but only one from the 10-excreta pools. Mosquito 21 followed the expectation that more excreta spots would result in more virus detection—the pool of 5 with positive DENV2 detection was at the upper limit of detection, with another pool classified as equivocal, while all three 10-spot pools exhibited lower Ct scores in comparison. Mosquito 19 was the only sample to have all six excreta pools test positive, although the number of spots in each pool seemed to make no discernible difference to the Ct score. Mosquito 19 excreta pools exhibited a wide range of Ct values as seen in the single spot experiment, with Ct scores ranging from 26.31 to 32.34 for the five-spot samples. This again is a ~100-fold difference in viral load between samples from the same mosquito and demonstrates the high variability whether using single or multiple excreta spots.

When whole mosquito Ct values exceeded 19, DENV2 detection in excreta became inconsistent. Mosquitoes 32 (Ct = 19.34), 9 (Ct = 20.89), and 17 (Ct = 22.87) returned three, one, and two positive pools, respectively, with more positive hits from ten-spot pools than from five-spot pools. Mosquito 8 (Ct = 34.32) was selected to test the ability of a whole mosquito with a virus load at the upper limit of positive to excrete virus at detectable levels using this experimental set-up. We tested pools of both 10 and 20 spots without successful detection of the virus, concluding that a very low mosquito viral load could not be compensated for with high excreta volume and reiterating the importance of mosquito viral load in excreta shedding.

With virus detection exhibiting high intra-mosquito and inter-mosquito variability, we investigated the minimum number of excreta spots required to repeatedly and reliably detect virus. For this experiment, mosquito excreta from multiple mosquitoes with different Ct scores was pooled into groups of 20 or 50, with the number of spots contributed from each mosquito uniform across the three replicates tested (Table 3). DENV2 was detected in two of the three 20-spot pool replicates, while the third replicate returned a borderline positive result. Positive Ct scores varied between 28.56 and 31.96, indicating a 100-fold difference in virus levels between the two replicates. The third replicate, which returned a borderline positive result, was again 100-fold and 10,000-fold lower than the first two. All three 50-spot pool replicates had detectable levels of DENV2, with Ct scores within a 0.19 range. Since it is likely that not all 50 spots included in these pool replicates contained DENV2 at detectable levels, we likely observed a diluting phenomenon caused by high numbers of negative excreta spots averaging out the positive spots. The threshold for reliable and reproducible detection therefore lies between 20 and 50 excreta spots.

### 3.4. Magnetic Concentration Expands Detection Capabilities

Since the pooling of excreta from multiple mosquitoes increased detection reliability and reproducibility, we sought to collect larger quantities of mosquito excreta into a single sample for virus detection. Collection and dissolution of over 50 excreta spots on dissolvable paper is impractical, and sample viscosity would likely hinder efficient RNA extraction. To overcome this, we collected excreta from DENV2-infected mosquitoes into PBS-A. Mosquitoes were grouped into pools of 5, 10, or 20 mosquitoes and housed in paper cups with a mesh bottom, suspended over a plastic cup containing 5 mL of PBS-A. Excreta could easily fall into the collection liquid, without the risk of contamination from dead mosquitoes or large debris (Figure 2a). Mosquito pools were homogenised, and the approximate DENV2 viral load of the pool was calculated. We assumed an infection rate of 66%, as observed previously for this virus. For this experiment, in addition to reporting the Ct score, we calculated DENV2 copies/µL from the Ct score using a DENV2 cDNA standard curve to allow for direct comparison of virus concentration between samples. Viral concentrations across the pools were approximately 1.57 × 10^4^ copies/µL (Ct = 19.89), 1.84 × 10^5^ copies/µL (Ct = 16.35), and 1.46 × 10^5^ copies/µL (Ct = 16.70) for the 5-, 10-, and 20-mosquito pools, respectively, confirming DENV2-positive mosquitoes in all pools and an approximate increase in total viral concentration as pool size increased.

Although liquid collection allows for easier accumulation of high excreta volumes, the excreta and therefore virus are unavoidably diluted. Dilution of excreta results in a reduction in virus concentration to levels below that of detection limits for the selected techniques, leading to potential false negative reporting and limitations on suitable detection techniques. We therefore trialled the use of magnetic beads (Mag4C Magnetic Beads; OZ Biosciences) for concentration of virus prior to testing with qPCR.

Of the 5 mL collection volume, 100 µL was reserved prior to magnetic concentration to serve as an ‘unconcentrated’ starting sample. The remaining volume was magnetically concentrated and resuspended in 100 µL buffer. The unconcentrated samples each reported average Ct scores ± SEM of 29.85 ± 0.92 (24.84 ± 8.71 copies/µL), 31.55 ± 0.84 (7.46 ± 2.73 copies/µL), and 31.44 ± 0.81 (7.85 ± 2.71 copies/µL) for the 5-, 10-, and 20-mosquito pools, respectively (Figure 2b). This result was surprising as we expected to see an increase in virus concentration as mosquito pool size also increased, whereas the opposite was observed. Following magnetic concentration, concentrated samples gave average Ct scores ± SEM of 29.14 ± 0.91 (45.89 ± 24.98 copies/µL), 27.97 ± 0.59 (77.40 ± 33.02 copies/µL), and 24.96 ± 0.43 (541.5 ± 173.4 copies/µL) for the 5-, 10-, and 20-mosquito pools, respectively (Figure 2b). Standard error was high for this experiment due to the variability in virus load between replicates in each of the pools tested. The 20-mosquito pool had the highest virus quantity and concentration factor of the three mosquito pools, with an average concentration factor of 119 ± 54.07 and varying from 31-fold to 249-fold between replicates. Concentration of virus was less effective as the number of mosquitoes in each of the tested pools decreased; the five- and ten-mosquito pools had average respective concentration factors of 1.92 ± 0.53 and 13.87 ± 4.48. Further, this demonstrated that by concentrating all virus within the sample, the quantity of virus can be more accurately determined.

## 4. Discussion

Surveillance and monitoring of arboviruses in Australia is currently limited to the manual collection of mosquitoes and extraction of viral material for PCR assays from whole mosquito grinds [4]. In this study, we assessed the suitability and feasibility of using *Ae. aegypti* excreta as an alternative sample type in DENV2 surveillance.

Initial experiments to evaluate virus detectability in mosquito excreta used dissolvable paper as a collection substrate. We selected dissolvable paper in favour of previously used collection surfaces such as parafilm [21,22], FTA cards [22], and filter paper [19,20] to allow for easy excision of individual excreta spots, no surface elution step or soaking, and assurance that an entire excreta spot was included within a sample without excess collection material. Testing of multiple individual excreta spots from different mosquitoes with similar viral loads demonstrated the high level of variability within and between mosquitoes. The proportion of positive excreta from a single mosquito, ranging from 10% to 100% depending on the mosquito selected, offered insight as to why we were unable to observe a clear correlation between spot number and Ct scores in the later pooled excreta experiments. Investigating the cause of variability was beyond the scope of this study, although it is likely dependent on several factors. Mosquito excreta was collected at 7–10 dpi to allow for adequate viral replication and dissemination [28] and to ensure that any virus detected within excreta was the result of infection and not simply due to passing of the spiked blood meal. However, the excreta collected over this three-day period were not separated according to the dpi at which they occurred, meaning any variation in excreta viral load between days could not be investigated. DENV1 viral load in the excreta of *Ae. aegypti* fluctuates over time after establishing infection [19], a phenomenon that likely extends to DENV2 infections. Further, all mosquito excreta samples were collected over the same time period, assuming an ideal collection window when, in actuality, ideal collection windows for viral detection in excreta are likely highly individual to each mosquito. We also did not account for variations in excreta volume, the possibility of RNA or virus breakdown within the experiment, or investigate viral dissemination within the mosquito itself. These were not within the scope of this pilot study, although they would be an informative route of investigation to further understand virus dynamics in mosquito excreta. Fontaine and coworkers similarly observed DENV1-negative excreta from DENV1-positive *Ae. aegypti* mosquitoes, again highlighting the individual nature of viral infection and excretion dynamics within mosquito populations [19].

Collecting DENV2-infected excreta from a pool of mosquitoes into a single liquid collection reservoir allowed for more convenient viral detection than the use of dissolvable paper. A liquid collection reservoir also lends itself to a more streamlined sample preparation, including magnetic concentration for higher accuracy viral load quantification. This was clearly observed as each of the 5-, 10-, and 20-mosquito pools appeared to have similar viral loads before concentration. Following concentration, the amount of virus detected in each sample increased as the mosquito pool increased, providing an accurate representation of the amount of virus truly present within a sample and not relying on diluted samples for quantification. The increase in virus quantification accuracy with the use of magnetic beads would likely address the high variability observed between samples within this study. If translating to a field surveillance setting, magnetic concentration would ensure that variability between viral excreta loads does not impact the testing reliability and that differences in results are reflective of true virus prevalence instead of detection limitations. This is the first published account of using Mag4C-Lv beads for the concentration of DENV2 virus. These beads are specifically designed for the concentration and assisted transfection of enveloped retroviruses, specifically lentiviruses, and have been used in such protocols to establish stable cell lines for immunological study purposes [29,30]. The beads have also been successfully used with viruses outside the Retroviridae family, with the isolation and concentration of viruses from the Togaviridae and Flaviviridae families [31,32]. Galasso and coworkers successfully used Mag4C-Lv beads to purify Zika virus, a flavivirus, and Mayaro virus, an alphavirus, bypassing the need for traditional sucrose gradient virus purification methods, which are time- and resource-consuming [32]. Mag4C-Lv beads were also utilised to isolate labelled Zika virus prior to mice infection studies [31]. Mag4C-Lv beads function through electrostatic and hydrophobic interactions between the magnetic nanoparticles and viral envelope proteins [33]. The flavivirus outer particle layer is primarily composed of the envelope (E), which has 40% amino acid sequence identity across the family [34]. This means that although the beads have only been shown to work with DENV2 and Zika virus to date, the beads may have further application for use with other mosquito-borne flaviviruses of concern, such as other DENV serotypes, WNV, and MVEV.

Magnetic concentration of excreta-derived virus could further be applied to laboratory-based vector competence and virus dissemination studies. The non-destructive technique allows for the detection of trace amounts of virus and can be used with samples collected across multiple time points. Ramirez and coworkers compared viral RNA levels between excreta and saliva collected from RRV-infected *Ae. vigilax* and WNV (Kunjin strain)-infected *Cx. annulirostris.* In both experiments, a higher proportion of excreta samples tested positive for viral RNA than saliva samples, either from a true lack of virus in salivary glands or due to very low expectoration volumes resulting in RNA material in the collection medium at levels below qPCR detection limits [21]. Here, magnetic concentration could be used to consolidate all expectorated virus for testing instead of using a fraction of an already low concentration sample and risking a false negative due to detection limits. Concentration further allows expansion into detection techniques that may have been precluded when using diluted samples with virus levels below the limit of detection, such as LAMP and immunoassay techniques. In-field LAMP has a wide application range and, amongst many examples, has been used to monitor arboviral vector presence [35,36], detect environmental contamination [37], medically diagnose [38], and evaluate food safety [39]. Next-generation arbovirus surveillance strategies are embracing the power of molecular detection and investigating how these highly sensitive techniques can be integrated into the current surveillance landscape. The in-field use of a mobile sequencing device for the detection of West Nile Virus from a single mosquito has been demonstrated [40], while Inglis and coworkers used in-field PCR to detect MVEV, RRV, and BFV also from whole mosquito grinds [41]. The use of mosquito excretion as a feasible sample type, combined with magnetic concentration to enhance detectability, aligns well with the current research aimed at mobilising molecular detection techniques for surveillance purposes.

Through this study we successfully demonstrated the detectability of DENV2 from the excreta of infected *Ae. aegypti* mosquitoes. While a virus could be detected from a single excreta spot using qPCR, the high degree of variability in excreta virus levels meant pooling of excreta together into a single sample increased the reliability of virus detection. Further, collection into a liquid substrate coupled with magnetic concentration streamlined excreta collection, facilitated more accurate virus quantification, and expanded available detection techniques. Complementary studies with a focus on in-field sample collection and adaptation of the current laboratory-based detection system would be advantageous for translation to field-based testing. Mosquito excreta is easy to collect, is readily generated by mosquitoes in reasonable quantities, and can be easily manipulated and applied to a range of different detection applications. These characteristics are advantageous in comparison to whole mosquito grinds or saliva, and as a surveillance sample, it adds to the toolbox of available strategies for arbovirus surveillance and monitoring.

## Figures and Tables

**Figure 1 pathogens-14-00042-f001:**
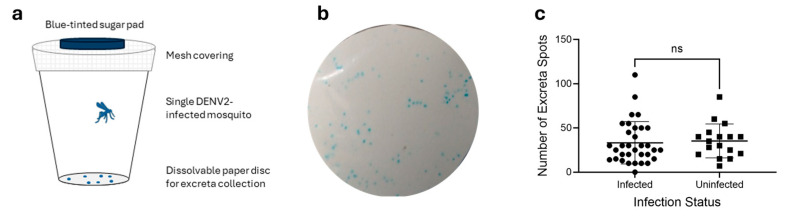
Excreta patterns of *Aedes aegypti* mosquitoes infected with DENV2 and uninfected. (**a**) DENV2-infected mosquitoes were housed individually in paper cups lined with dissolvable paper and supplied with blue-tinted sugar pads. (**b**) Representative figure of *Ae. aegypti* mosquito excreta after consumption of 10% sucrose supplemented with blue food dye. (**c**) Number of total excreta spots per mosquito from 7 dpi to 10 dpi. Average excreta spots per mosquito were 33.2 and 35.4 for infected and uninfected pools, respectively. Statistical significance calculated using Prism GraphPad (v9.1.2) Student’s *t*-test. ns indicates a *p* value > 0.05 or not significant.

**Figure 2 pathogens-14-00042-f002:**
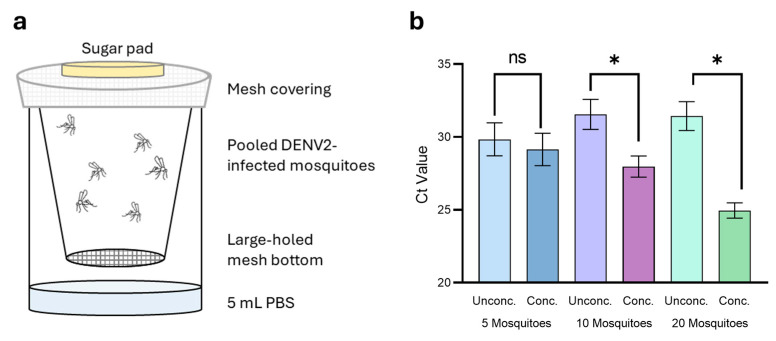
Liquid collection and magnetic concentration of DENV2 from *Ae. aegypti* excreta. (**a**) DENV2-infected mosquitoes were housed in pools of 5, 10, and 20 in plastic cups suspended over 5 mL PBS-A. (**b**) Excreta collected from pools of 5 (blue), 10 (purple), and 20 (green) mosquitoes were concentrated using Mag4C-Lv magnetic beads. Statistical significance calculated using Prism GraphPad (v9.1.2) paired *t*-test, * indicates *p* value < 0.05, ns indicates a *p* value > 0.05 or not significant.

**Table 1 pathogens-14-00042-t001:** qPCR of single excreta spots from DENV2-infected *Aedes aegypti* mosquitoes. Five mosquitoes of similar Ct value and excreta numbers were selected, and viral load of ten single spots from each mosquito were tested using qPCR.

Mosquito ID	Positive	Equivocal	Negative	Average Ct ± SEM	Positive Ct Range
Mosquito 5 (Ct = 17.03)	10	0	0	30.29 ± 0.80	27.45–34.79
Mosquito 44 (Ct = 17.21)	6	0	4	32.39 ± 1.08	27.10–34.22
Mosquito 42 (Ct = 17.59)	2	3	5	33.25 ± 01.19	31.57–34.94
Mosquito 1 (Ct = 17.62)	1	0	9	35.30 ± 0	35.30
Mosquito 28 (Ct = 17.72)	7	3	0	32.44 ± 0.78	28.99–35.39

Positive = Ct score below 36.0; Negative = Ct score above 36.0 or undetermined; Equivocal = one positive replicate.

**Table 2 pathogens-14-00042-t002:** Pooling of excreta spots from DENV2-infected *Aedes aegypti* mosquitoes. Six mosquitoes with viral load Ct values ranging from 16.72 to 22.87 were selected, and excreta spots pooled into samples including five or ten spots each, with each pool size repeated in triplicate.

Mosquito ID	No. Spots per Pool	Positive Pools	Equivocal Pools	Negative Pools	Average Ct ± SEM	Positive Ct Range
Mosquito 4 (Ct = 16.72)	5 10	3 0	0 1	0 2	30.96 ± 1.02 ND	29.15–32.70 ND
Mosquito 21 (Ct = 17.52)	5 10	1 3	1 0	1 0	34.28 ± 0 31.15 ± 0.12	34.28 30.91–31.11
Mosquito 19 (Ct = 17.87)	5 10	3 3	0 0	0 0	29.25 ± 1.74 29.54 ± 1.49	26.31–32.34 26.78–31.88
Mosquito 32 (Ct = 19.34)	5 10	1 0	0 2	2 1	31.86 ± 0 ND	31.86 ND
Mosquito 9 (Ct = 20.89)	5 10	0 0	0 1	3 2	ND ND	ND ND
Mosquito 17 (Ct = 22.87)	5 10	1 0	0 1	2 2	33.63 ± 0 ND	33.63 ND

Positive = Ct score below 36.0; Negative = Ct score above 36.0 or undetermined; Equivocal = one positive replicate.

**Table 3 pathogens-14-00042-t003:** Pooling of 20 and 50 mosquito excreta spots from DENV2-infected *Aedes aegypti* mosquitoes.

No. of Spots	Replicate	Pool 1 Ct	No. of Spots	Replicate	Pool 2 Ct
20	1	28.56	50	1	32.18
	2	34.61 *		2	32.10
	3	31.96		3	32.37

* One duplicate tested positive.

## Data Availability

All data supporting the findings of this study are available within this paper.

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
