# Peer review of "Exploring Mosquito Excreta as an Alternative Sample Type for Improving Arbovirus Surveillance in Australia"

_pathogens, 2025, doi:10.3390/pathogens14010042_

Round 1
Reviewer 1 Report
Comments and Suggestions for Authors
The manuscript entitled "Exploring mosquito excreta as an alternative sample type for improving arbovirus surveillance in Australia" by Malcolm et al, is a good report on applying new strategies for epidemiological surveillance of vector-borne diseases. The manuscript is well written. The methodology is clear, and the results are interesting. However, this strategy is not novel and has been tested by several groups. On the other hand, the collection of viruses in liquid substrate is interesting, but its application in the field is complicated and time-consuming. It is essential to test both strategies in the field and correlate the results with human cases. In this way, surveillance may impact the prevention of VBD.
Author Response
Comment 1: The manuscript entitled "Exploring mosquito excreta as an alternative sample type for improving arbovirus surveillance in Australia" by Malcolm et al, is a good report on applying new strategies for epidemiological surveillance of vector-borne diseases. The manuscript is well written. The methodology is clear, and the results are interesting. However, this strategy is not novel and has been tested by several groups. On the other hand, the collection of viruses in liquid substrate is interesting, but its application in the field is complicated and time-consuming. It is essential to test both strategies in the field and correlate the results with human cases. In this way, surveillance may impact the prevention of VBD.
Response 1: We would like to thank the reviewer for their comments and agree that in-field testing of liquid substrate collection would provide valuable information on the logistics of this collection technique, but is beyond the scope of the current study. This study was designed as a pilot, researching alternative methods to sample collection than what is currently available and demonstrates that collection into liquid substrate allows for easier downstream manipulation of the sample (i.e. magnetic concentration to enhance detection capabilities) and a more streamlined pipeline for testing methods that require liquid sample inputs than collection onto a solid surface. We would also like to highlight that excreta collection into liquid substrate, in combination with magnetic concentration has potential to improve workflows in several laboratory applications, including vector competence studies, vertical transmission studies and viral excreta viability studies, and is not limited to field studies only.
Reviewer 2 Report
Comments and Suggestions for Authors
The authors investigate the detection of DENV-2 virus from mosquito excreta spots under laboratory conditions. While in the study all mosquitoes were fed with DENV-2, there is a significant challenge in translating this technique to field applications. In natural populations, the prevalence of DENV-infected mosquitoes is typically low, raising concerns about the practicality of pooling samples, as this would likely dilute the viral concentration and reduce detection sensitivity. Additionally, given that RNA viruses degrade over time in environmental conditions, it would be useful to understand the correlation between detection rates and environmental exposure duration.
As a methodological study, a critical issue is the lack of clarity regarding experimental design. Key experimental details, such as sample sizes, the number of replicates, and specific protocols, are not provided.
How are excreta spots collected, and how is environmental exposure duration controlled? Prolonged exposure could increase the likelihood of false negatives due to RNA degradation.
How many mosquitoes were used to generate data for single excreta spots (Table 1)? Similarly, what was the sample size for pooled excreta spot experiments (Table 2)? When examining pooled excreta spots, how did the authors determine the specific pool sizes (e.g., 5, 10, 20, or 50)?
To enhance the reliability and applicability of the method, significant revisions are required. The authors should conduct experiments simulating field conditions where DENV-2 prevalence is low to validate the pooling strategy and detection limits. Additionally, studies investigating the impact of environmental exposure on RNA stability over varying time intervals are essential to establish the method’s robustness in real-world scenarios. Without these experiments and a clearer presentation of the experimental design, the study lacks the necessary foundation to support its field applicability.
Author Response
Comment 1: The authors investigate the detection of DENV-2 virus from mosquito excreta spots under laboratory conditions. While in the study all mosquitoes were fed with DENV-2, there is a significant challenge in translating this technique to field applications. In natural populations, the prevalence of DENV-infected mosquitoes is typically low, raising concerns about the practicality of pooling samples, as this would likely dilute the viral concentration and reduce detection sensitivity.
Response 1: We would like to thank the reviewer for this comment. During this study we present the use of magnetic beads for virus concentration and demonstrate how these may be used in scenarios where virus prevalence is low. As this is a pilot study testing proof of principle, future studies would be required to investigate sensitivity of the detection system using virus at levels similar to those found in the field. Further investigations would also involve adaptation of the system to detect viruses that are commonly found in Australia, such as Ross River virus or Murray Valley Encephalitis virus, followed by possible translation to field application.
Comment 2: Additionally, given that RNA viruses degrade over time in environmental conditions, it would be useful to understand the correlation between detection rates and environmental exposure duration.
Response 2: We have identified RNA degradation as a limitation of this study and acknowledge that excreta collected on different day post exposure may have experienced varying levels of RNA degradation. These limitations are outlined in the conclusion and highlighted as a further study area.
Comment 3: As a methodological study, a critical issue is the lack of clarity regarding experimental design. Key experimental details, such as sample sizes, the number of replicates, and specific protocols, are not provided.
Response 3: We have amended the description of mosquito sample size on page 4, paragraph 1, line 154 to read “Mosquito excreta was collected from 50 mosquitoes between 7 and 10 dpi on dissolvable…” to clarify total number of mosquitoes used for solid phase collection experiments. The following statement has been added to clarify replicates for each experiment and batch size used: “For single spot experiments, 10 spots from a single mosquito were excised and processed individually as 10 separate replicates. For batches of 5 and 10, excreta spots were excised from a single mosquito and pooled into three replicate pools of 5 and three replicate pools of 10. For batch sizes of 20 and 50, excreta spots were excised in equal numbers from five different mosquitoes and pooled into three replicate pools of 20 spots and three replicate pools of 50 spots” on page 4, paragraph 1, lines 162-168.
All protocols and methodologies have been described in the Materials and Methods section, with commercially available kit protocols referenced where they have been used.
Comment 4: How are excreta spots collected, and how is environmental exposure duration controlled? Prolonged exposure could increase the likelihood of false negatives due to RNA degradation.
Response 4: Collection of excreta spots onto dissolvable paper is described in the methods section, page 4, paragraph 1 lines 155-157, with a schematic depicting the housing and collection set-up in Figure 1a. Excision of excreta spots from dissolvable paper and method of dissolving paper in PBS is described on page 4, paragraph 1, lines 160-161. Collection of excreta into the liquid PBS substrate is described in the methods section, page 4, paragraph 2, lines 178-181, with a schematic depicting the housing and collection set-up in Figure 2a.
Environmental exposure duration is controlled insofar as all mosquito excreta samples were collected between days 7 – 10 post exposure. We outline the limitations of collecting excreta for a fixed period of time in the discussion (page 10, paragraph 1, lines 375-381), in that excreta virus load may vary within the 7-10 days post exposure and that viral RNA in excreta deposited earlier may undergo degradation in comparison to those deposited later.
Comment 5: How many mosquitoes were used to generate data for single excreta spots (Table 1)?
Response 5: Five mosquitoes were included in the single excreta spot experiment. The number of mosquitoes and the selection criteria used to select the mosquitoes is outlined in page 5, paragraph 2 lines 239-242. Reference to Table 1 has been included in page 5, paragraph 2, line 241 to clarify. Table 1 heading has been updated from “Table 1. qPCR of single excreta spots from DENV2-infected Aedes aegypti mosquitoes” to “Table 1. qPCR of single excreta spots from DENV2-infected Aedes aegypti mosquitoes. 5 mosquitoes of similar Ct value and excreta numbers were selected and viral load of 10 single spots from each mosquito tested using qPCR.”
Comment 6: Similarly, what was the sample size for pooled excreta spot experiments (Table 2)?
Response 6: We have updated the text on page 6, paragraph 1, line 247 to read “…experiment, spots from six individual mosquitoes were pooled into batches of 5 or 10 and…”, clarifying that six mosquitoes were used for this experiment. The selection criteria for mosquito selection are outlined on page 6, paragraph3, lines 259-260.
Comment 7: When examining pooled excreta spots, how did the authors determine the specific pool sizes (e.g., 5, 10, 20, or 50)?
Response 7: Pool sizes were selected to represent low, medium and high amounts of excreta that may be expected from a single mosquito. Over the three-day collection period, the average amount of excreta spots deposited per mosquito was 34.0, with a range of 0 – 108. These figures were used as the basis for testing different pool sizes.
Comment 8: To enhance the reliability and applicability of the method, significant revisions are required. The authors should conduct experiments simulating field conditions where DENV-2 prevalence is low to validate the pooling strategy and detection limits. Additionally, studies investigating the impact of environmental exposure on RNA stability over varying time intervals are essential to establish the method’s robustness in real-world scenarios. Without these experiments and a clearer presentation of the experimental design, the study lacks the necessary foundation to support its field applicability.
Response 8: We thank the author for this comment. As this study was a pilot project testing the feasibility of excreta collection onto both solid substrate and into liquid substrate coupled with magnetic concentration, field studies are/or simulations are outside the scope of the work presented within in this manuscript. We are interested in conducting studies on the degradation of viral particles and viral RNA in environmental conditions in the future, although would conduct this as a stand-alone study with a focus on in-field translation rather than laboratory-based methods.
The following statement has been added to the conclusion paragraph, highlighting that field studies or field-like studies would need to take place for the process of adaptation to in-field monitoring to occur: “Complementary studies with a focus on in-field sample collection and adaptation of the current laboratory-based detection system would be advantageous for translation to field-based testing” on page 11, paragraph 2, lines 445 – 447.
Reviewer 3 Report
Comments and Suggestions for Authors
The manuscript "Exploring mosquito excreta as an alternative sample type for improving arbovirus surveillance in Australia" is current and relevant, especially given the need to improve arbovirus surveillance in tropical and subtropical regions. The text is clearly written, but there are gaps in the contextualization and some redundancies throughout the manuscript.
______________________________________________________________________
Methodology
There is not enough description about the use of dissolvable paper, preparation of solutions and pooling criteria. Therefore, it is necessary to add technical details (thickness of dissolvable paper, type of PBS used).
Line: 154-155. The same information has already been described previously. It is not necessary to insert it again.
Line: 153-163. The text is repetitive, as it mentions the methodologies above. The suggestion would be to insert the “new” information together with the paragraph “Breeding, raising and infection of Aedes aegypti”.
Only the t-test was used. More robust statistical methods would be adequate to deal with high variability. It is recommended to consider ANOVA or regression models to compare groups more rigorously.
The description of magnetic concentration does not include parallel tests for other control samples.
Results
Line 335- Put the word Aedes aegypti in italics.
Figures and tables are useful, but some do not have sufficiently informative captions. Captions need to be improved (e.g.: Tables 1 and 2 need to detail the experimental conditions better).
The inconsistent detection of DENV2 in excreta is not sufficiently explored. It is recommended to analyze in more depth the reasons for negative results, even in infected mosquitoes.
Ct thresholds used for positivity could be discussed in greater depth, especially considering variability.
Data on the impacts of pooling (e.g. virus dilution) could be presented in more informative graphs.
Discussion
It is necessary to include studies on cost-benefit and applicability of alternative methods.
High variability in results is mentioned, but without practical implications. It is necessary to discuss how variability may impact field surveillance.
Conclusion
Statements such as "has the potential to revolutionize arbovirus surveillance" are too broad without field validation. It is suggested to limit conclusions to the context of the experiment and suggest complementary studies.
________________________________________
Statements such as "there was no external funding" are appropriate, but it would be useful to detail the indirect funding institutions.

Author Response
Comment 1: There is not enough description about the use of dissolvable paper, preparation of solutions and pooling criteria. Therefore, it is necessary to add technical details (thickness of dissolvable paper, type of PBS used).
Response 1: The following information was added with additional details of dissolvable paper: “A lightweight fabric work stabilizer (Sulky, Solvy water soluble stabiliser), with comparable thickness to printer paper was used to create the dissolvable paper discs.” on page 4, paragraph 1, lines 156-157. The PBS type and formulation have been added to the methods section on page 3, paragraph 3, lines 118-120 and PBS amended to PBS-A throughout the manuscript.
Comment 2: Line: 154-155. The same information has already been described previously. It is not necessary to insert it again.
Line: 153-163. The text is repetitive, as it mentions the methodologies above. The suggestion would be to insert the “new” information together with the paragraph “Breeding, raising and infection of Aedes aegypti”.
Response 2: To clarify the different mosquito conditions used between experiments, the methods have been amended in the Breeding, raising and infection’ sections to read: “mosquitoes selected at random and placed in in paper cups designed for excreta collection onto a solid surface (dissolvable paper) or into a liquid substrate (PBS-A; 137 mM NaCl, 2.7 mM KCl, 8 mM Na2HPO4 and 2 mM KH2HPO4, pH 7.4). Mosquitoes were provided with paper towel soaked with 10% w/v sucrose solution placed on top of the cup. The sucrose solution was spiked with blue food dye according to experimental requirements. At 10 dpi, whole mosquitoes were destroyed by freezing for 20 min then collected into 200 µL Qiagen RLT Plus buffer (Qiagen) and stored at -80°C until use. Dissolvable paper or PBS-A was collected from each cup and stored in individual plastic bags or tubes at -80°C until further use.” On page 3, paragraph 3, lines 118 – 126.
This means the Breeding section is more general and applicable to both solid and liquid substrate collection, while eliminating repetition of conditions between methods sections.
Comment 3: Only the t-test was used. More robust statistical methods would be adequate to deal with high variability. It is recommended to consider ANOVA or regression models to compare groups more rigorously.
Response 3: We’d like to thank the reviewer for their comment and suggesting we reevaluate the statistical methods used to compare virus levels before and after magnetic concentration. We have done so and believe the paired t-test is the most applicable method for statistical analysis, as we are comparing one sample before and after an intervention/treatment. We have updated the methods section, and Figure 2 to reflect the use of the paired t-test in place of the t-test.
Comment 4: The description of magnetic concentration does not include parallel tests for other control samples.
Response 4: We would like to thank the reviewer for their comment and ask for clarification about what is meant by ‘other control samples’?
Comment 5: Line 335- Put the word Aedes aegypti in italics.
Response 5: Aedes aegypti has been italicised.
Comment 6: Figures and tables are useful, but some do not have sufficiently informative captions. Captions need to be improved (e.g.: Tables 1 and 2 need to detail the experimental conditions better).
Response 6: Table 1 heading has been updated from “Table 1. qPCR of single excreta spots from DENV2-infected Aedes aegypti mosquitoes” to “Table 1. qPCR of single excreta spots from DENV2-infected Aedes aegypti mosquitoes. 5 mosquitoes of similar Ct value and excreta numbers were selected and viral load of 10 single spots from each mosquito tested using qPCR.”
Table 2 heading has been updated to read: “Table 2. Pooling of excreta spots from DENV2-infected Aedes aegypti mosquitoes. 6 mosquitoes with viral load Ct values ranging from 16.72 to 22.87 were selected and excreta spots pooled into samples including 5 or 10 spots each, with each pool size repeated in triplicate.”
Comment 7: The inconsistent detection of DENV2 in excreta is not sufficiently explored. It is recommended to analyze in more depth the reasons for negative results, even in infected mosquitoes.
Response 7: Thank you for this comment, this is an area that we are interested in exploring further, although was outside the scope of this study. We describe some probable causes of variability in our discussion section, including mosquito infection time frames and potential RNA degradation throughout the collection period depending on when the sample was deposited. Clarifying the reason for differences in viral load between excreta spots would likely involve in-depth time-point monitored excreta experiments, coupled with histological examination of the infected mosquitoes to understand the systemic infection state, which is beyond the scope of the current study.
Comment 8: Ct thresholds used for positivity could be discussed in greater depth, especially considering variability.
Response 8: We have added in the following statement explaining the Ct threshold cut off for positive excreta samples: “Each mosquito excreta sample was tested using qPCR in technical duplicate as described above and deemed positive if Ct values were below 36. A higher Ct cut-off was selected compared to the whole mosquito experiments to reflect the reduction in material used for testing” on page 4, paragraph 1, lines 170-173.
While several samples recorded Ct scores between 35-36, we did not observe any successful qPCR results with Ct scores above 36.0.
Comment 9: Data on the impacts of pooling (e.g. virus dilution) could be presented in more informative graphs.
Response 9: Thank you for this suggestion to present the Table 3 data in graphs instead of a table. When presenting this data as a graph, important nuanced details are lost that are necessary to interpret the results in a non-misleading fashion. We therefore prefer to keep the data presented in a table format.
Comment 10: It is necessary to include studies on cost-benefit and applicability of alternative methods.
Response 10: Because this is a pilot project, we would not foresee that the sample collection and combined magnetic concentration methods would be deployed in its current iteration, and do not see the need for cost-benefit analysis to current sample collection methods. We have also described the applicability of the protocols used in this manuscript to laboratory processes such as vertical transmission studies or vector competence studies.
Comment 11: High variability in results is mentioned, but without practical implications. It is necessary to discuss how variability may impact field surveillance.
Response 11: Thank you for this comment. We have added the following short statement into the discussion to address this comment:
“The increase in virus quantification accuracy with the use of magnetic beads would likely address the high variability observed between samples within this study. If translating to a field surveillance setting, magnetic concentration would ensure that variability between viral excreta loads does not impact the testing reliability and that differences in results are reflective of true virus prevalence instead of detection limitations.” On page 10, paragraph 2, lines 393 – 398.
We would also like to highlight that the protocols presented within this manuscript have applicability beyond field surveillance and could be optimised for laboratory experiments such as vector competence studies.
Comment 12: Statements such as "has the potential to revolutionize arbovirus surveillance" are too broad without field validation. It is suggested to limit conclusions to the context of the experiment and suggest complementary studies.
Response 12: This statement is in the final sentence of the abstract. We have amended this to read “has the potential to improve arbovirus surveillance” to more suitably reflect the outcomes of the study.
Comment 13: Statements such as "there was no external funding" are appropriate, but it would be useful to detail the indirect funding institutions.
Response 13: We have added the following comment to reflect funding sources: “This research was internally funded by Commonwealth Scientific and Industrial Research Organization (CSIRO).”
Round 2
Reviewer 2 Report
Comments and Suggestions for Authors
The authors have addressed all comments.